# The Spectroscopic Atomic and Molecular Databases at the Paris Observatory

**Evelyne Roueff** [1,*] ![ID], **Sylvie Sahal-Bréchot** [1,†] ![ID], **Milan S. Dimitrijević** [1,2,†] ![ID], **Nicolas Moreau** [1,†] ![ID] **and Hervé Abgrall** [1,†] ![ID]

1   Observatoire de Paris, PSL University, Sorbonne Université, CNRS, LERMA, 92190 Meudon, France;
    sylvie.sahal-brechot@obspm.fr (S.S.-B.); mdimitrijevic@aob.rs (M.S.D.); nicolas.moreau@obspm.fr (N.M.);
    herve.abgrall@obspm.fr (H.A.)
2   Astronomical Observatory, Volgina 7, 11060 Belgrade, Serbia
*   Correspondence: evelyne.roueff@obspm.fr
†   These authors contributed equally to this work.

**Abstract:** This paper is intended to give a comprehensive overview of the current status and developments of the Paris Observatory STARK-B, MOLAT and SESAM databases which can be interrogated thanks to interoperability tools. The STARK-B database provides shifting and broadening parameters of different atomic and ionic transitions due to impacts with charged particles (the so-called Stark broadening) for different temperatures and densities. The spectroscopic MOLAT and SESAM databases provide the wavelengths, the oscillator strengths or Einstein spontaneous emission coefficients of $H_2$, CO and isotopologues molecules.

**Keywords:** atomic and molecular physics; stark broadening; UV spectra

## 1. Introduction

Astrophysical environments can only be probed remotely thanks to particle or photon messengers collected through space or ground-based telescopes. The analysis of the recorded spectra is expected to allow for derivation of the physical conditions of the observed regions. However, this can only be achieved if the fundamental basic microscopic processes involving atoms, ions, molecules, electrons, etc., are well enough identified and understood. We describe in this review the efforts developed at the Paris Observatory to provide complete and relevant spectral information dedicated to astrophysical and laboratory plasmas and colder molecular environments through online databases. Section 2 reports the STARK-B database providing the shifting and broadening parameters of different atomic and ionic transitions due to electron impact, referred to as Stark broadening, for different temperatures and densities. We present in Section 3 the spectroscopic MOLAT and SESAM databases which provide the wavelengths and the transition strengths of different molecules.These achievements have greatly benefited from the interoperability tools developed within the VAMDC project which will be described in another paper of this issue by M.L. Dubernet and coworkers. The integration process of SESAM and STARK-B databases into the VAMDC architecture and the software tools that were used are described in Section 4.

## 2. STARK-B

Interpretation of collisional broadening and shifting of spectral lines is important for providing information on the medium, such as temperatures; densities of the perturbers; and chemical abundances of the emitting or absorbing atoms or molecules. Due to the great and continuous progress of astrophysical observations and laboratory measurements, accurate spectroscopic diagnostics are

more and more needed. Modeling of synthetic spectra requires the knowledge of numerous spectral line profiles, which can be modified by collisions with the particles of the medium, especially electron and positive ions for hot and medium-hot stellar atmospheres and their interiors. For laboratory plasmas, the increasing development of magnetic confinement fusion devices (Tokamaks, such as WEST (formerly Tore Supra) or the ITER project) of inertial confinement fusion devices (such as NIF or "Laser MégaJoule"), and technological plasmas also needs many atomic data, including line profiles due to interactions with electrons and ions of the medium. The access to such data via online databases has become indispensable.

In Astrophysics, the interest for databases emerged at the IAU XXIInd General Assembly (1994), where a Joint discussion "Atomic Databases, New power Source for Astronomy" was organized by W. L. Wiese [1].

In fact, since the eighties, Sahal-Bréchot, Dimitrijević and coworkers have been updating and operating at a large scale the numerical code that was created by Sahal-Bréchot several years earlier for calculating Stark broadening and shifting of isolated lines in the impact complete collision approximation of neutral and ionized elements. Though this broadening mechanism is usually called "Stark" broadening, it is a matter of collisions with electrons and ions and not of Stark effect. This code, named SCP, is based on the impact semi-classical-perturbation approximation [2]. The range of considered temperatures and densities is large, and depends on the degree of ionization of the studied element.

A summary of that theory and the approximations can be found in [2] and earlier papers cited in [2]. The numerical calculations are very fast. Thus, the method is especially adapted for obtaining a great number of data in the same run. Moreover, it is accurate enough for the needs of users: 20–30% on average for the widths. The SCP results agree with the other theoretical methods [3–5], and with experimental results, [5] for instance. The uncertainty of the SCP results is due to the use of the perturbation theory [2]. For the shifts, the results can sometimes be less accurate, due to negative interference effects between the upper and lower levels of the studied line.

Thanks to this collaborative work, which is currently being continued, more than 150 papers, concerning 125 neutral and ionized elements broadened and shifted by collisions with electrons and ions, have been published.

In addition, the development of the Virtual Observatory, due to the great amount of data which needed to be connected, increased in the beginning of this new century. Interoperable e-infrastructures were urgently needed. This request enabled the creation of databases at the Paris Observatory.

Then LERMA[1] laboratory gave us a support to put all the results of our numerous tables in a database named STARK-B [6]. A precursor, named BELDATA, was foreseen by AOB (Astronomical Observatory of Belgrade) [7]. Then, STARK-B started at Paris Observatory, with S. Sahal-Bréchot and M.S. Dimitrijević, who developed the database with the support of LERMA, as part of the databases of the Paris Observatory. STARK-B opened on line at fall 2008 with a link to the Serbian Virtual Observatory [8]. Currently, S. Sahal-Bréchot and M.S. Dimitrijević are the scientists co-responsible for the database, and N. Moreau is responsible for all the technical aspects. STARK-B is in free access [6] and is also accessible through the thumbnail of the Scientific databases of the website of Paris Observatory [9], in addition to MOLAT [10].

In 2009, the European project VAMDC (Virtual Atomic and Molecular Data Centre) [11], was born, and this helped us a lot. STARK-B became a node of VAMDC. VAMDC is a secure, documented, flexible, interoperable platform based on e-science, which permits optimized search and exchange of atomic and molecular data. The STARK-B data model follows the VAMDC standards, and STARK-B is also accessible via the VAMDC portal [12].

---

[1]　　refer to the Abbreviations Section

STARK-B is aimed at meeting the above cited needs for widths and shifts of isolated lines of neutral and ionized atoms due to electron and ion collisions (i.e., impacts). The elements of interest are numerous. The ranges of temperatures and densities are large, and depend on the degree of ionization of the studied element: for the temperatures from 2500 K to about $6 \times 10^6$ K, and for the electron densities from $10^{10}$ to $10^{23}$ cm$^{-3}$.

Presently, STARK-B contains Stark broadening and shifts for many transitions calculated with the SCP method. Besides the SCP results, it also contains a number of MSE (modified semi-empirical) datums obtained by Dimitrijević, Konjević and coworkers [13–15]. This method is more approximate (estimated averaged error bar ±40%) than the SCP method but can be used for emitters when atomic data are not sufficiently complete to perform a semiclassical perturbation calculation. This is often the case for heavy elements.

## 2.1. Description of STARK-B Database

Since 2010, approximately every two years the database and its detailed content has been presented in several workshops and conferences. We will only give a brief outline in the following.

The homepage of STARK-B (http://stark-b.obspm.fr) has several menus. "Introduction" summarizes the theoretical methods and approximations, and provides references to the basic papers. "Data Description" describes the data that are in the files. "Access to the Data" provides a graphical interface with the periodic table with access for getting to the required data. The user is guided throughout the query toward the desired element; the required ionization degree; and the required lines, densities, temperatures and colliding perturbers. Concerning SCP data, colliding perturbers are electrons, and most often protons; He$^+$ and He$^{++}$ ions, which interest stellar physics; and frequently, other positive ions interesting for laboratory or technological plasmas. For MSE data, only widths (and sometimes shifts) due to electron collisions exist. The user can make a query by domain of wavelengths or by transitions. The data model (in particular the identification of the levels of the line transitions) follows the VAMDC standards, in order to allow interoperability with other atomic databases included in VAMDC. So the transitions are defined by configurations, terms and J-values, but different couplings can be taken into account. Then a table displaying the widths and the shifts is generated. According to the theory, when LS coupling is valid, the data are given for multiplets only and not for the individual fine structure lines. A number of warnings (validity of impact approximation, of isolated line approximation, of complete collision approximation) are included in the tables. They are commented on in "Introduction" and "Data Description" menus. Bibliographic references to the papers are provided, which can be freely downloaded if access is not restricted. Then widths and shifts data can be downloaded in ASCII and in VOTable format (XML format) and adapted to Virtual Observatory tools.

For the computer codes leading to the modeling of stellar atmospheres, fitting formulae and coefficients as functions of temperature for every line are asked for, because they are much more convenient for the numerical calculations than widths and shifts tabulated for a set of temperatures. Consequently, an ad-hoc simple and accurate fitting formula using the Levenberg–Marquardt algorithm for least-squares method has been obtained for interpolation in the displayed data [16]. Thus another table has been added to each table of widths and shifts. It gives the fitting coefficients with the temperature, which can also be downloaded in ASCII and in VOTable format, and introduced in the computer codes for stellar modeling.

Newly added datasets and revised datasets are displayed in the "Data History" menu. The "Contact" menu is intended for further inquiries and user support. Almost 2000 queries have been performed on the STARK-B website over 2018 and 2019. The access query through VAMDC is minor (less than one hundred per year).

*2.2. Future and Conclusions*

In the short and medium term, we will continue to implement our published calculated data in STARK-B. With our coworkers, we also continue to carry on new calculations of line widths and shifts for needs in modeling and spectroscopic diagnostics. Furthermore, we plan to implement our published quantum results in STARK-B to develop other fitting formulae, such as regularities and systematic trends, and to include them in STARK-B.

## 3. MOLAT and SESAM

The MOLAT [10] and SESAM [17] databases provide atomic and molecular data computed or gathered by members of the Paris Observatory and some other spectroscopic databases. The consulting and possible downloads are fully accessible without any restriction. The focus is principally on UV electronic spectroscopy, which reflects the availability of the 10 m VUV spectrograph on the Meudon site of the Paris Observatory and the work performed on the LURE synchrotron radiation facility on the Orsay Campus of Paris University. These experimental studies were closely linked to several spatial missions: the 1200–1500 Å spectral range was indeed available with the International UV Explorer (IUE) and the Hubble Space Telescope (HST) with the Goddard High Resolution Spectrograph (GHRS). In the early 2000s, the 912–1200 Å spectral range became accessible with the far UV Spectroscopic Explorer (FUSE).

*3.1. MOLAT*

The MOLAT database was set up about thirty years ago. The deliveries are essentially ascii files with proper references. Nevertheless, some data are also displayed in PDF format. We focus here on the data which have been subsequently translated in the XML language via the XSAMS protocol to comply with interoperability requirements. They correspond to VUV Spectra of $H_2$ and its isotopologues and CO. The specificity and we think the interest of the deliveries are the joined information of spectral transition wavenumbers/wavelengths and the corresponding oscillator strengths or Einstein emission transition probabilities. Most spectroscopic experimental papers report accurate transition wavenumbers which are subsequently interpreted in terms of Hamiltonian properties with derivation of term energy values. The corresponding transition oscillator strengths require additional experimental (lifetime studies) or theoretical and computational work involving ab-initio quantum chemistry calculations of the interatomic potentials, possible couplings and transition moments followed by the integration of the matrix elements of the transition moment over the radial variable.

3.1.1. CO

Two different spectral ranges are displayed.

- The spectrum from the intersystem transitions of CO, involving a′ $^3\Sigma^+$-X$^1\Sigma^+$, e $^3\Sigma^+$-X$^1\Sigma^+$ and d$^3\Delta$-X$^1\Sigma^+$ is reported in [18]. The corresponding transitions take place in the $\sim$ 1250–1650 Å window, where the so-called fourth positive system A$^1\Pi$-X$^1\Sigma$ transitions are also available but are often subject to large opacity effects in absorption astronomical spectra.
- The VUV spectrum between 911.2 and 1152.2 Å of CO and different isotopomers is also displayed, following [19]. These transitions involve 31 different electronic states and are sometimes subject to predissociation, resulting from the numerous possible couplings between nearby electronic states. These results are of paramount importance to computing photodissociation of CO under interstellar conditions and possible isotopic selective photodissociation effects.

3.1.2. $H_2$ VUV Spectrum

The data reported for $H_2$ are the results of a long term collaboration between experimentalists and theoreticians [20,21]. The data are displayed for four electronic systems, B$^1\Sigma_u^+$-X$^1\Sigma_g^+$ (Lyman band system), C$^1\Pi_u$-X$^1\Sigma_g^+$ (Werner band system), B′$^1\Sigma_u^+$-X$^1\Sigma_g^+$ and D$^1\Pi_u$-X$^1\Sigma_g^+$. The four upper electronic

states are coupled through both radial and rotational terms, and the theoretical computations taking this into account allow one to derive the spectroscopic properties of the different electronic transitions. The emission probabilities towards the continuum of the ground state, i.e., above the dissociation energy of $H_2$, are also reported up to a rotational quantum number J = 25. These data have been extensively used by astrophysicists for different applications (interpretation of FUSE observations of VUV absorption transitions [22], far UV emission from circumstellar disks [23,24], emission bump in protoplanetary disks [25], auroral and dayglow emission of Uranus atmosphere [26], Jovian auroral spectroscopy [27]). The remarkable accuracy of the reported computed wavelength is a fraction of an Angstrom. This achievement results from the semi-empirical approach applied in the calculations [28], where the energy terms of the ground electronic state are the experimental values of [29] and the upper electronic potential functions have been slightly and carefully modified compared to the ab-initio values [30] to reproduce experimental transition wavelengths, when available.

*3.2. SESAM*

The SESAM database represents a recent initiative aiming to update the results displayed in MOLAT and offers new tools based on the VAMDC project. The present SESAM database is restricted to the VUV spectrum $H_2$ and its isotopes through its Lyman $B^1\Sigma_u$-$X^1\Sigma_g$, Werner $C^1\Pi_u$-$X^1\Sigma_g$ , $B'^1\Sigma_u$-$X^1\Sigma_g$ and $D^1\Pi_u$-$X^1\Sigma_g$ electronic transitions as well as the CO spectrum, corresponding to the $A^1\Pi$-$X^1\Sigma$ system (sometimes called the fourth positive system) and the close perturbing $e^3\Pi$-X and $a^3\Sigma$-X systems. The present status offers the possibility to search the transitions of $H_2$, HD, $D_2$ and CO in specific wavelength ranges, as displayed in Table 1. The original data have been written in the XML format as defined by the XSAMS group [31] and allow one to fully recover the identifications of the quantum numbers involved in the transitions, the wavelengths (in Å), the transition wavenumbers $\sigma$ in reciprocal centimeters and the oscillator strengths $f$. That information allows for the computation of the emission transition probabilities ($A$ in $s^{-1}$) from the following formula: $A_{u \to l} = \frac{2\pi e^2 \sigma^2}{mc\epsilon_0} \cdot \frac{g_l}{g_u} \cdot f_{lu}$, where $u/l$ refer to the upper/lower states and $g_u$, $g_l$ are the corresponding statistical weights. When $\sigma$ is expressed in reciprocal centimeters, the emission probability ($s^{-1}$) is obtained from $A_{u \to l} = 0.667 \cdot \sigma^2 \cdot \frac{g_l}{g_u} \cdot f_{lu}$.

**Table 1.** Wavelength range (Å) of $H_2$, HD, $D_2$ and CO consulting data.

| Molecule | Minimum Value | Maximum Value |
|----------|---------------|---------------|
| $H_2$ | 844.76 | 1844.57 |
| HD | 747.33 | 1852.03 |
| $D_2$ | 745.15 | 1855.8 |
| CO | 1173.16 | 1568.89 |

The complete source files are also downloadable in ascii format, allowing the users to process them with their favorite software. The number of queries of SESAM via the web has been rather limited up to now, about hundred per year, whereas those dedicated to MOLAT reached more than five hundred queries per year since 2016. The access via VAMDC was not available until recently.

A number of improvements are foreseen for SESAM, resulting from the recent achievements obtained in theoretical studies of the various corrections (non-adiabatic, relativistic, radiative) introduced in the $H_2$, HD and $D_2$ potential curves [32] as well as the amazing spectral resolution of electronic spectra obtained thanks to vacuum-ultraviolet Fourier transform spectrometers put in front of synchrotron radiation sources [33,34]. Whereas previous data remain useful, it is time to redistribute the data while taking into account recent work. Particular care will be put into the references to original work. The improved accuracy available on the wavelength measurements [35,36] represents also a major step for the investigation of fundamental constant variations in observations of redshifted absorption transitions detected towards a quasar background source. The VUV spectra are shifted

towards the visible region of the spectrum thanks to the relation $\lambda_{obs} = \lambda_{vacuum} \times (1 + z)$ where $z$ is the redshift. A tool allowing one to compute the spectra at any redshift will be made available as well.

## 4. Interoperability

### 4.1. VAMDC Environment

The VAMDC infrastructure is based on a set of standards that, together, provide the interoperability layers. The corner stone of the system is the data model, called XSAMS (XML schema for atomic, molecular and solid data [37]), which is now officially endorsed by the IAEA. It covers many fields of atomic and molecular physics and provides a common definition of the quantities to the data provider. This schema are used at different levels. First and foremost, being an XML schema, it defines the structure of an XML document. Instances of these schema are used as data exchange formats between the elements of the infrastructures. They are also used to define the names of queryable and searchable quantities in the query language vocabulary. VAMDC also defined an access protocol to query the VAMDC compatible databases. It provides a common query language allowing a user to query many services at the same time. With each one returning the result in the same XSAMS format, the user can use the same tools to analyze them. The VAMDC-compatible databases are called VAMDC nodes. They must be compatible with the standards. In order to make their implementation simpler, data producers can use the VAMDC node-software [38], a python middleware that can be connected to the database, taking care of understanding the request and generating the XSAMS documents. All the VAMDC nodes are registered into a service registry. It stores the capabilities of each node (which are the queryable parameters and returned quantities). It can be searched by a user but also by software, which can discover existing services according to the quantities it returns for example. The VAMDC registry relies on the standard defined by the International Virtual Observatory Alliance for the implementation of resource registries [39].

### 4.2. Implementation on the Databases

As implementing standards is a tedious process, VAMDC provides a middleware application, called the node software, taking care of it. This python application, based on the Django framework, is able to understand the user queries, translate them into SQL so that the underlying database can execute them and generate a XSAMS document from the results. The main condition to be able to use the node software is that the database relies on a relational database management system, such as MySQL, MariaDB or PostgreSQL.

Since its origin, STARK-B relied on MySQL. Integrating it into VAMDC implied few modifications in its schema, mainly adding new mandatory information that was not available at the time (like the InchI/InchIKey description of chemical species [40]).

From the SESAM point of view, work began with VAMDC in mind from the ground up. Consequently, the database structure is quite similar to the XSAMS structure. The Figure 1 presents the lowest level of this structure. At the center of the database is the transition. It has a relation with the upper and the lower level. Each one of them is linked to a chemical species (identified by its InchIKey).

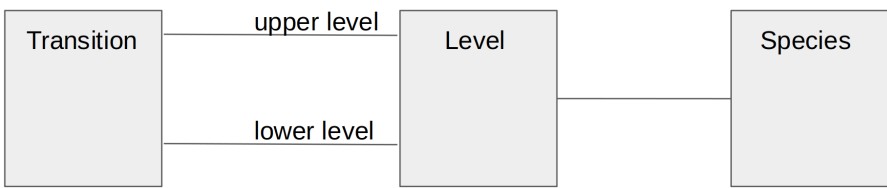

**Figure 1.** Core structure of the databases.

The other parts of the databases have nothing in common as they deal with very different data. Once it has been properly configured, the node software execute the queries against the content of those tables. In Appendices A and B, tables listing the queryable and returned quantities for both nodes are available.

The configuration of the middleware consists of declaring in a file which columns of the database correspond to standardized one in the XSAMS data model. It can be done in a matter of hours which ensures a quick deployment, if the data are already stored in a relational database system.

**Author Contributions:** E.R. coordinated the full paper. The draft of Section 2 was prepared by S.S.-B. and M.S.D.; that of Section 3 by E.R. and H.A.; and that of Section 4 by N.M. All authors have read and agreed to the published version of the manuscript.

**Funding:** This research received support from VAMDC. VAMDC is funded under the Combination of Collaborative Projects and Coordination and Support Actions Funding Scheme of The Seventh Framework Program. Call topic: INFRA-2008-1.2.2 Scientific Data Infrastructure. Grant agreement number: 239108.

**Acknowledgments:** This work has also been supported by the Paris Observatory (LERMA and Paris Astronomical Data Centre (PADC)), the CNRS and the Programme National de Physique Stellaire (INSU-CNRS). The cooperation agreements between Tunisia (DGRS) and France (CNRS) (project code 09/R 13.03, number 22637) are also acknowledged. The support of IAEA is gratefully acknowledged. The support of Ministry of Education, Science and Technological Development of the Republic of Serbia through projects 176002 and III44022 is also acknowledged.

**Conflicts of Interest:** The authors declare no conflict of interest. The funders had no role in the design of the study; in the collection, analyses or interpretation of data; in the writing of the manuscript, or in the decision to publish the results.

## Abbreviations

The following abbreviations are used in this manuscript:

| | |
|---|---|
| ASCII | American Standard Code for Information Interchange |
| FUSE | Far Ultraviolet Space Explorer |
| GHRS | Goddard High Resolution Spectrograph |
| HST | Hubble Space Telescope |
| ITER | the Way in latin |
| IUE | International Ultraviolet Explorer |
| LERMA | Laboratoire d'Etudes du Rayonnement et de la Matière en Astrophysique |
| LURE | Laboratoire d'Utilisation du Rayonnement Electromagnétique |
| MSE | Modified Semi- Empirical code |
| NIF | National Ignition Facility |
| SCP | Semi-Classical Perturbation code |
| SESAM | SpEctroScopic Atomic and Molecular database |
| VAMDC | Virtual Atomic and Molecular Data Center |
| VUV | Vacuum Ultra Violet |
| WEST | Tungsten (W) Environment in Steady-state Tokamak |
| XML | Extensible Markup Language |
| XSAMS | XML Schema for Atomic, Molecular and Solid Data |

## Appendix A. STARK-B Service Queryable and Returnable Quantities

**Table A1.** STARK-B service.

| Returnables | Restrictables |
|---|---|
| EnvironmentTotalNumberDensity | EnvironmentTotalNumberDensity |
| AtomStateTermLSL | InchiKey |
| AtomSpeciesID | RadTransWavelength |
| AtomStateTermJKS | EnvironmentTemperature |
| NodeID | AtomSymbol |
| AtomNuclearCharge | IonCharge |

**Table A1.** *Cont.*

| Returnables | Restrictables |
|---|---|
| RadTransBroadeningPressureLineshapeParameterUnit | |
| AtomStateTermJKK | |
| AtomSymbol | |
| AtomStateTermLSS | |
| AtomInchiKey | |
| RadTransBroadeningPressure | |
| RadTransWavelengthUnit | |
| AtomStateRef | |
| AtomStateParity | |
| EnvironmentTotalNumberDensityUnit | |
| EnvironmentTemperatureUnit | |
| RadTransUpperStateRef | |
| SourceURI | |
| AtomStateTermLSMultiplicity | |
| SourceName | |
| SourceYear | |
| AtomIonCharge | |
| SourceCategory | |
| RadTransShiftingEnv | |
| EnvironmentSpeciesName | |
| RadTransBroadeningPressureEnvironment | |
| EnvironmentID | |
| RadTransWavelength | |
| ParticleSpeciesID | |
| RadTransSpeciesRef | |
| SourceAuthorName | |
| AtomStateTermLabel | |
| RadTransID | |
| EnvironmentSpeciesRef | |
| AtomStateTotalAngMom | |
| RadTransBroadeningPressureLineshapeParameterComment | |
| AtomMassNumber | |
| SourceVolume | |
| RadTransBroadeningPressureLineshapeParameter | |
| RadTransRefs | |
| RadTransShiftingParamName | |
| SourceTitle | |
| AtomInchi | |
| RadTransShiftingParam | |
| RadTransShiftingParamUnit | |
| EnvironmentTemperature | |
| SourceID | |
| AtomStateTermJKJ | |
| AtomStateID | |
| ParticleName | |
| RadTransBroadeningPressureLineshapeParameterName | |
| AtomStateConfigurationLabel | |
| RadTransBroadeningPressureLineshapeName | |
| RadTransShiftingName | |
| RadTransLowerStateRef | |

## Appendix B. SESAM Service Queryable and Returnable Quantities

**Table A2.** SESAM service quantities.

| Returnables | Restrictables |
|---|---|
| MoleculeQNSpinComponentLabel | InchiKey |
| RadTransID | RadTransWavenumber |
| SourceDOI | RadTransWavelength |
| MoleculeQNr | RadTransProbabilityOscillatorStrength |
| MoleculeStateID | MoleculeStoichiometricFormula |
| MoleculeInchiKey | MoleculeChemicalName |
| MoleculeQNv | StateEnergy |
| RadTransWavelength | RadTransProbabilityA |
| MoleculeOrdinaryStructuralFormula | |
| MoleculeQNelecInv | |
| RadTransWavelengthUnit | |
| RadTransWavenumberMethod | |
| MoleculeQnCase | |
| RadTransUpperStateRef | |
| MoleculeQNF | |
| SourceID | |
| MoleculeQNJ | |
| MoleculeQNasSym | |
| MoleculeQNN | |
| MoleculeStoichiometricFormula | |
| MoleculeQNS | |
| RadTransWavenumberUnit | |
| SourceCategory | |
| MoleculeStateTotalStatisticalWeight | |
| MoleculeQNLambda | |
| RadTransProbabilityA | |
| MoleculeInchi | |
| MethodCategory | |
| MethodID | |
| RadTransSpeciesRef | |
| SourceAuthorName | |
| RadTransProbabilityOscillatorStrength | |
| RadTransProbabilityAUnit | |
| SourceVolume | |
| RadTransRefs | |
| MoleculeQNparity | |
| RadTransWavenumber | |
| MoleculeQNelecRefl | |
| MoleculeStateEnergyOrigin | |
| NodeID | |
| MoleculeStateEnergyUnit | |
| SourceYear | |
| MoleculeQNKronigParity | |
| MoleculeStateEnergy | |
| SourceURI | |
| MoleculeSpeciesID | |
| RadTransWavenumberComment | |
| MoleculeChemicalName | |
| MoleculeQNF1 | |
| MoleculeQNElecStateLabel | |
| RadTransLowerStateRef | |

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
