# Peer review of "The Spectroscopic Atomic and Molecular Databases at the Paris Observatory"

_atoms, doi:10.3390/atoms8030036_

Round 1

Reviewer 1 Report

The authors have provided an overview of their databases located in Meudon, but linked into the wider system VAMDC.  The range of applicability is specified in the paper, and covers research and data for which the authors are internationally recognised as experts.  The data contained in this database is very important for astrophysical applications, and there is an ongoing need for further elements to be considered. The context of the special issue is an appropriate medium for this update of what is currently available with pointers to subsequent directions and extensions.  I assume that access to the database is via VAMDC, but it would be useful if the authors could say so explicitly in the text, to provide easy to follow pointers for new users.

The paper is clearly presented, but I have indicated that, in a few places, the grammar might be improved:

lines 51: 'enough accurate' would be better as 'accurate enough' or sufficiently accurate'.

lines 51-52: 'for the widths in average' would be better as 'on average for the widths'.

line 67: 'for' instead of 'of'.

line 74: 'to meet' would be better as 'at meeting'.

line 77: perhaps a multiplication sign could be added after '6'.

lines 186 and 201: 'to compute' would be better as 'the computation of'.

line 231: replace 'moment' by 'time'?

Author Response

We thank the referee for his (her) detailed comments. We have complied with all his (her) suggestions in the revised version. The updates have been written in boldface.

Reviewer 2 Report

The paper describes three databases in atomic/ molecular physics, with interest for astrophysics. 

The advantage of this paper is to recall these databases, even if they are, rather limited. Therefore, I ask adjustments in the paper and some critical checking of the databases by authors.

Major questions:

Authors should explain better in the text, what is the difference (advantage) of one database over another. SESAM seems to contain the same data as MOLAT (if not, explain it), and furthermore in a less direct (zipped) format.

A detailed description of the three databases, that were organized by authors, could serve also to adjust/ update some details in these databases. Some examples:

  1. Stark home page says “in the impact approximation”. A reference is needed already here.

In the “introduction”, it is said that the two approximations have rather big uncertainty bars. The most recent reference (Popovic) is from 1996. Do we know which phenomena make the uncertainty bar so large?

  1. MOLAT seems be a copy of a book published in 1994. Some check-reading would be needed: page https://molat.obspm.fr/index.php?page=pages/Atlas/intro.php B2Su+ state is once written with subscripts, once without.

Reference that are quoted on that page should be given at the bottom.  

3. "Stark" can be accessed also from MOLAT: the paper should say this.

4. Is there a way to find these databases from the main page of https://www.observatoiredeparis.psl.eu/?lang=fr?

5. Excessive self-citations: remove 14-17.

6. Finally, authors should mention some statistics on accesses to the described databases, even if low.

Minor changes:

Line 16 should read: online

Line 34 remove double parenthesis

Explain acronyms WEST and  NIF

Line 51 in the same run

Line 77 should read 6x10^6

Line 187 separate the formula from the text

Author Response

We thank the referee for his (her) detailed remarks. 

We have updated the introduction homepage of the H2 UV Atlas at http://molat.obspm.fr/index.php?page=pages/Atlas/intro.php and added the corresponding references as requested.

The references for the impact approximation  are also reported at https//stark-b.obspm.fr/index.php/introduction.

We have also complied with all his (her) suggestions, adding the references, pointing to the addresses where the databases are accessible and giving the statistics of the queries. Self-citations (previously 14-17) have been removed. 

As explained in the paper, SeSAM is a recent initiative aiming to update the results  reported in MOLAT, when available and providing new tools resulting from the VAMDC project such as the queries on line of specific molecular transitions within a wavelength range.

All the modifications and updates are written in boldface to ease the checks of the referee.